# KIR+ CD8+ T Lymphocytes in Cancer Immunosurveillance and Patient Survival: Gene Expression Profiling

**DOI:** 10.3390/cancers12102991

**Published:** 2020-10-15

**Authors:** Lourdes Gimeno, Emilio M. Serrano-López, José A. Campillo, María A. Cánovas-Zapata, Omar S. Acuña, Francisco García-Cózar, María V. Martínez-Sánchez, María D. Martínez-Hernández, María F. Soto-Ramírez, Pedro López-Cubillana, Jorge Martínez-Escribano, Jerónimo Martínez-García, Senena Corbalan-García, María R. Álvarez-López, Alfredo Minguela

**Affiliations:** 1Immunology Service, Hospital Clínico Universitario Virgen de la Arrixaca and Instituto Murciano de Investigación Biomédica (IMIB-Arrixaca), 30120 Murcia, Spain or lgarias@um.es (L.G.); josea.campillo@carm.es (J.A.C.); angiecanovas@gmail.com (M.A.C.-Z.); mariav.martinez3@carm.es (M.V.M.-S.); mariad.martinez85@carm.es (M.D.M.-H.); mariaf.soto@carm.es (M.F.S.-R.); mdrocio.alvarez@gmail.com (M.R.Á.-L.); 2Human Anatomy Department, University of Murcia (UM), 30100 Murcia, Spain; 3Department of Biochemistry and Molecular Biology, Veterinary School, International Excellence Campus “Campus Mare Nostrum”, Universidad Murcia, 30100 Murcia, Spain; emiliomanuel.serrano@um.es (E.M.S.-L.); senena@um.es (S.C.-G.); 4Biomembranes and Cell Signaling, Instituto Murciano de Investigación Biomédica, 30120 Murcia, Spain; 5Faculty of Veterinary and Zootechnics, Autonomous University of Sinaloa (UAS), Culiacan 80246, Mexico; osacunam@uas.edu.mx or; 6Department of Research, Animal Reproduction Biotechnology (ARBiotech), Culiacan 80015, Mexico; 7Department of Biomedicine, Biotechnology and Public Health (Immunology), University of Cadiz and Institute of Biomedical Research Cádiz (INIBICA), 11009 Cadiz, Spain; curro.garcia@uca.es; 8Urology Service, Hospital Clínico Universitario Virgen de la Arrixaca and Instituto Murciano de Investigación Biomédica (IMIB-Arrixaca), 30120 Murcia, Spain; pedrolopezcubillana@gmail.com or; 9Dermatology Service, Hospital Clínico Universitario Virgen de la Arrixaca and Instituto Murciano de Investigación Biomédica (IMIB-Arrixaca), 30120 Murcia, Spain; jorge.mescribano@gmail.com; 10Oncology Service, Hospital Clínico Universitario Virgen de la Arrixaca and Instituto Murciano de Investigación Biomédica (IMIB-Arrixaca), 30120 Murcia, Spain; jeronimo@seom.org; 11Faculty of Health Sciences, Universidad Católica San Antonio de Murcia, UCAM, 30107 Murcia, Spain

**Keywords:** cancer, CD8+ T cells, KIR, immunosurveillance, gene expression, microarray

## Abstract

**Simple Summary:**

Killer-cell immunoglobulin-like receptors (KIR) are molecules expressed by the most important cells of the immune system for cancer immune vigilance, natural killer (NK) and effector T cells. In this manuscript we study the role that cytotoxic CD8+ T cells expressing KIR receptors could play in cancer immune surveillance. With this objective, frequencies of different KIR+ CD8+ T cell subsets are correlated with the overall survival of patients with melanoma, ovarian and bladder carcinomas. In addition, the gene expression profile of KIR+ CD8+ T cell subsets related to the survival of patients is studied with the aim of discovering new therapeutic targets, so that the outcome of patients with cancer can be improved.

**Abstract:**

Killer-cell immunoglobulin-like receptors (KIR) are expressed by natural killer (NK) and effector T cells. Although KIR+ T cells accumulate in oncologic patients, their role in cancer immune response remains elusive. This study explored the role of KIR+CD8+ T cells in cancer immunosurveillance by analyzing their frequency at diagnosis in the blood of 249 patients (80 melanomas, 80 bladder cancers, and 89 ovarian cancers), their relationship with overall survival (OS) of patients, and their gene expression profiles. KIR2DL1+ CD8+ T cells expanded in the presence of HLA-C2-ligands in patients who survived, but it did not in patients who died. In contrast, presence of HLA-C1-ligands was associated with dose-dependent expansions of KIR2DL2/S2+ CD8+ T cells and with shorter OS. KIR interactions with their specific ligands profoundly impacted CD8+ T cell expression profiles, involving multiple signaling pathways, effector functions, the secretome, and consequently, the cellular microenvironment, which could impact their cancer immunosurveillance capacities. KIR2DL1/S1+ CD8+ T cells showed a gene expression signature related to efficient tumor immunosurveillance, whereas KIR2DL2/L3/S2+CD8+ T cells showed transcriptomic profiles related to suppressive anti-tumor responses. These results could be the basis for the discovery of new therapeutic targets so that the outcome of patients with cancer can be improved.

## 1. Introduction

The tumor microenvironment induces the suppression of the immune response by means of multiple mechanisms [1]. However, therapy-induced tumor burden reduction may allow the immune system to recover, thus offering a second chance to some patients. In fact, after the initial therapy, it is common to observe highly variable survival periods, even in patients with tumors with similar histopathological and genetic characteristics, which suggests the participation of factors other than the tumor itself. The immunogenetic background of patients is perhaps the factor with the greatest impact on cancer susceptibility, response to tumor therapies, and patient survival [2]. Thus, it has been described that the presence of lymphoid cells infiltrating the tumor microenvironment correlates with better patient prognosis [3].Tumors actively contribute to modulating the microenvironment by secreting factors such as the platelet-derived growth factor (PDGF)-DD, which can support tumor growth and stromal reaction, but concomitantly activates innate immune responses. NKp44, which is encoded by the NCR2 gene and expressed on activated natural killer (NK) cells, innate lymphoid cells, and gamma/delta T cells, recognizes PDGF-DD and triggersthe secretion of IFN-γ and TNF-α, which induce tumor cell growth arrest [4]. Other receptors, such as NKG2D, play decisive roles in tumor immune response. NKG2D is a potent activating immunoreceptor expressed on nearly all cytotoxic lymphocytesthat promotes cytotoxicity against cells expressing NKG2Dligands (MICA/B and ULBP1-6), which are induced bycellular stress, viral infection, or malignant transformation [5]. Experimental mice models have shown that NKG2D/ligand interactions on NK cells can induce a strong secretion of IL-2 and IFN-γ,which contribute to suppressing tumor growth by modulating tumor microenvironment [6].

Cytotoxic CD8+ T lymphocytes and Natural Killer (NK) cells have different mechanisms of target recognition and signaling cascades, but both cell subsets play pivotal roles in tumor immunosurveillance [7]. Naïve CD8+ T cells infiltrating the tumor differentiate into effector and memory CD8+ T cells in order to perform their targeted functions at the tumor site. However, while it is generally admitted that CD8+ T cells are directly involved in antitumor cytotoxic responses, the role of CD8+ T cells expressing killer-cell immunoglobulin-like receptors (KIR) in cancer immune surveillance remains elusive [8]. Like NK cells, effector memory KIR+ CD8+ T cells are cytotoxic and can produce IFN-γ [9]. The expression of KIR receptors on CD8+ T cells correlate with advanced maturation stages as evidenced by a memory phenotype (negative expression of CCR7 and CD28), high perforin content, reduced proliferative potential, and poor IFN-γ production upon TCR engagement [9,10,11]. Apparently, KIR receptors regulate specific T cell effector functions by modulating TCR signaling pathways [12,13], which might represent a protective mechanism to control potentially harmful cytolytic CD8+ T cells, by raising their activation threshold [11]. Different reports have evidenced that KIR+ CD8+ T lymphocytes were expanded in patients with tumors due to chronic antigenic stimulation [14]. However, more recent results described that, at early stages of tumor development, tumor-induced IFNγ-producing KIR+ CD25+ CD127− FOXP3− CD8+ T cells can potentiate immune surveillance by targeting human leukocyte antigen (HLA)-E-restricted CD4+ regulatory T cells (Treg) while leaving the effector T cell population unaffected [15]. At a later stage of tumor development, when CD4+ Treg cells dominate the tumor-microenvironment, CD4+ Tregs mediate the clonal deletion of these tumor-suppressive KIR+ IFNγ+ CD25+ CD8+ T cells and ensure tumor immune evasion. Besides, at later stages of tumor development a FOXP3+ TGFβ+ CD25+ CD127− CD8+ T cell population is expanded [15].

The effector function of CD8+ T and NK cells is regulated by the balance between the activating and inhibitory signals delivered by a broad variety of receptors [10,13], among them the KIR receptors, originally described on NK cells [16], but also expressed in minor subsets of peripheral blood T cells [9,10,11]. KIR receptors and their HLA class-I (HLA-I) ligands are highly diverse. In fact, diversity in the repertoire of KIR/HLA-ligand interactions determines the susceptibility to autoimmunity, infections, or cancer (http://www.allelefrequencies.net/tools/kirDiseaseBib.aspx) [16]. Depending on the KIR genotype, each individual may express different numbers and combinations outof 9 inhibitory KIR (iKIRs, KIR2DL1-4, KIR2DL5a/b and KIR3DL1-3) and 6 activating KIR (aKIRs, KIR2DS1-5 and KIR3DS1) receptors [17]. Although the role of iKIR/HLA-ligand interactions in CD8+ T lymphocytes is largely unknown, in NK cells they induce fully competent cells in a process known as “licensing”. The best characterized iKIR/HLA-I licensing interactions are KIR2DL1/HLA-C2, KIR2DL2/L3/HLA-C1 (although KIR2DL2 may also interact with HLA-C2), and KIR3DL1/HLA-Bw4 allotypes [8]. Interactions of aKIRs have also been reported between KIR2DS1/HLA-C2 allotypes, KIR2DS2/C1, KIR2DS2/HLA-A11, KIR2DS2/non-HLA cancer cell-expressed ligand, KIR2DS4/HLA-A*11 and a limited number of HLA-C1 and -C2 allotypes, KIR3DS1/HLA-F, and KIR3DS1*014/HLA-Bw4. The role of aKIRs in NK cell education is less well-known. In contrast to inhibitory receptors, aKIR signaling, specifically KIR2DS1 in HLA-C2 homozygous individuals, renders NKcells hyporesponsive, a mechanism that has possibly evolved to prevent autoreactivity. Apparently, aKIR/HLA-I interactions have detrimental effects on the tumor immune surveillance of NK cells, with aKIR richer B-haplotypes showing higher susceptibility to cancer and/or worse outcomes. In fact, higher frequency of aKIRs or Bx centromeric and telomeric genotypes has been associated with gastric cancer, non-Hodgkin lymphoma, and childhood acute lymphoblastic leukemia. However, these results contradict those reporting survival benefits for patients with acute myeloid leukemia (AML) who have received grafts from unrelated donors with 1 or 2 KIR B-haplotypes (recently reviewed by Guillamón et al. [18]). Although, aKIRs and NKG2D share DNAX activating proteins (DAP10 and DAP12) in the signaling pathwaysof T and NK cytotoxic cells [19], NKG2D signaling determines distinct functional outcomes: direct activation of cytotoxicity and cytokine production in NK cells, but co-stimulation in activated CD8+ T cells only, so that T cells will require additional signals, especially from the T cell receptor (TCR), to build up a complete functional response. For this reason, the co-participation of additional activating or inhibiting receptors, such as KIRs, could be of relevance in the anti-tumor response of CD8+ T cells. Nonetheless, the role of NKG2D and KIR co-signaling in CD8+ T lymphocytes, to our knowledge, remains unexplored.

The aim of this study was to determine the role of KIR+ CD8+ T cells in cancer immune surveillance by analyzing the frequency of these cells in peripheral blood of patients with different types of solid cancers at diagnosis, and their association with specific KIR/HLA-ligand interactions and patient survival. Additionally, gene expression analysis was conducted by DNAmicroarray in KIR2D+ and KIR2D− CD8+ T cells expanded in vitro from individuals with different genetic backgrounds, to characterize molecular pathways and immune response mechanisms acting in these cells, with the objective of identifying potential targets to enhance the anti-tumor activity of these cytotoxic cells.

## 2. Results

### 2.1. C2-Ligand Was Associated with Expansions of KIR2DL1+ CD8+ T Cells in Surviving Patients

No significant differences were observed between healthy controls and cancer patients regarding the frequency of major peripheral blood T cell subsets CD4+ (39.4% vs. 39.4%) and CD8+ (23.6% vs. 23.2%), total NK cells (13.4% vs. 13.8%) (Figure 1A), or CD8+ T cells expressing different KIR receptors, KIR2DL1+ (1.4% vs. 1.7%), KIR2DS1+ (0.4% vs. 0.36%), KIR2DL1/S1+ (0.04% vs. 0.04%), KIR3DL1+ (0.93% vs. 1.27%), KIR2DL2+ (3.05% vs. 2.16%), and KIR2DL3+ (3.82% vs. 4.37%) (Figure 1B). However, a significant expansion of KIR2DL1+ CD8+ T cells, but no expansion of other KIR+ CD8+ T cell subsets, was observed in cancer patients in the presence of their specific HLA-C2 ligand (2.2% vs. 0.84%, *p* = 0.009, C1C2/C2C2 vs. C1C1 patients) (Figure 1C), while no significant differences among KIR+ CD8+ T cell subsets were detected in the presence of HLA-Bw4 ligands (Figure 1D) or HLA-C1 ligands. Appendix A shows KIR+ CD8+ T cell repertoires breakdown for each type of solid cancer.

The expansion of KIR2DL1+ CD8+ T cells induced by its specific C2-ligand was observed in all cancer patients (2.21% vs. 0.94%, *p* < 0.01, compared to patients without the C2-ligand) and maintained in patients who survived the monitoring period(2.45% vs. 0.86%, *p* < 0.01), but abrogated in patients who died during the follow-up (0.62% vs. 0.61%) (Figure 2A). The expansion of KIR2DL1+ CD8+ T cells induced by its specific C2-ligand was observed in healthy controls and in patients who survived the monitoring periodwith any of the three types of cancer analyzed (Figure 2B).

### 2.2. C1-Ligand Was Associated with Dose-Dependent Expansion of KIR2DL2/S2+ CD8+ T Cells and Sorter Patient Survivals

Next, we analyzed the impact of the specific C1-ligand on the frequency of KIR2DL2/S2+ CD8+ T cells at diagnosis and its association with patient survival. In contrast to KIR2DL1+ CD8+ T cells, which were expanded specifically in patients who survived the monitoring period, KIR2DL2/S2+ CD8+ T cells were expanded in the presence of their specific C1-ligand (4.06% vs. 0.73%, *p* < 0.01) in patients who died during the follow-up (1.82% vs. 2.68%, compared with surviving patients) (Figure 2C). This expansion observed in patients who died during the follow-up was present in any of the three types of cancer analyzed (Figure 2D). In fact, the expansion of KIR2DL2/S2+ CD8+ T cells observed at diagnosis in patients who died during the follow-up was C1-ligand dose dependent (5,45%, 2,82%, and 0,73% for C1C1, C1C2, and C2C2, *p* < 0.01) (Figure 2E).

The impact of the C1-ligand on the overall survival (OS) of cancer patients was investigated with Kaplan-Meier analysis to show that the presence of C1-ligand was associated with shorter OS (87.7 vs. 93.6 months, *p* < 0.05). In contrast, the presence of the C2-ligand did not impact patient OS (Figure 2F).

### 2.3. KIR+ NK Cell Repertoires Were Not Associated to the Survival of Cancer Patients

Since the expression of KIR receptors is predominant in NK cells, an analysis of KIR+ NK cell repertoires wasperformed followinga similar strategy to the one used to analyze CD8+ T lymphocytes (Figure 3). As previously described in the patients of this series [20], KIR+ NK cell repertoires did not show significant differences between healthy controls and cancer patients (Figure 3A). Besides, the presence of HLA C1-ligands did not alter the distribution of KIR+ NK cell subsets. However, the presence of HLA C2-ligands was associated with a higher frequency of KIR2DL1+ NK cells (8.88 vs. 4.23, *p* < 0.01) (Figure 3B), a difference that persisted in all groups of cancer patients regardless of whether they survived or succumbed to the disease during follow-up (Figure 3C). In the same line, and in contrast to the results observed for KIR+ CD8+ T lymphocytes, the presence of HLA C1-ligands was not significantly associated with higheramounts of KIR2DL2/S2+ NK cells in patients who died during the follow-up (Figure 3C).

### 2.4. In Vitro, KIR2D+ CD8+ T Cells Expanded Preferentially in the Presence of HLA-C1 and IL-12

To gain further insight into the functional properties of KIR2D+ CD8+ T cells in cancer, we next analyzed and isolated these cells after in vitro expansion. To do so, PBMCs from healthy donors with three different HLA-C genotypes (C1C1, C1C2, and C2C2) (Figure 4) were stimulated in vitro with the PBMC of the same C1C2 donor. Whilst KIR2D-negative CD8+ T cells showed similar expansion kinetics for the three types of HLA-C genotypes (1327, 1131, and 1232 cells × 10^3^ at day 19, respectively), KIR2DL1/S1+ (14.5, 13.0, and 4.2 cells × 10^3^ at day 19, respectively) and KIR2DL2/L3/S2+ (114.1, 87.7, and 23.7 cells × 10^3^ at day 19, respectively) CD8+ T cell subsets were notably expanded in C1C1 and C1C2 genotypes, but much less expanded in the C2C2 genotype (Figure 4B). Therefore, although C1C1, C1C2, and C2C2 genotypes did not influence the fold change at day 19 of KIR2D− CD8+ T cells (11.1, 10.9, 10.0, respectively), they notably impacted that of KIR2DL1/S1+ (39.5, 20.4, and 5.9, respectively), and KIR2DL2/L3/S2+ (33.6, 20.9, and 11.7, respectively) CD8+ T cells (Figure 4C).

Next, we evaluated the influence of different cytokines in the expansion of KIR2D− and KIR2D+ CD8+ T cells (Figure 4C). The addition of IL-12 (5 ng/mL) to the standard protocol highly increased the expansion of KIR2DL1/S1+ CD8+ T cells (323.7 fold change), slightly increased the expansion of KIR2DL2/L3/S2+ CD8+ T cells (188 fold change), and reduced the expansion of KIR2D− CD8+ T cells (51.4 fold change), compared to the standard protocol (98.2, 150.7, and 121.8 fold change). IL-10 (20 ng/mL) did not alter the fold change of KIR2D−, KIR2DL1/S1+ and KIR2DL2/L3/S2+ CD8+ T cells (84.7, 146.3, and 94.7) at day 19. However, IL-15 (10 ng/mL; 68.5, 53.8, and 36.6 fold change), IFNγ (50 ng/mL; 32.2, 38.9, and 46.7 fold change), and mainly TGFβ (1 ng/mL; 25.1, 24.1 and 19.4 fold change) clearly suppressed the expansion of KIR2D−, KIR2DL1/S1+, and KIR2DL2/L3/S2+ CD8+ T cells (Figure 4D,E).

The proportion of perforin positive T cells remained unaltered after the in vitro expansion in standard conditions compared to their initial levels in both KIR2D− CD8+ (19.2% vs. 28.0%) and KIR2D+ CD8+ T cells (62.0% vs. 67.1%).

### 2.5. Transcriptional Profiling in KIR2D+ CD8+ T Cells: Functional Analysis

Molecular status of CD8+ T cells was assessed by using microarray gene expression performed in total RNA extracted from highly purified KIR2D− (purity > 99.5%) and KIR2D+ (purity > 98.0%) subsets after a 19-day in-vitro expansion. To analyze differences in gene expression profile between KIR2D− and KIR2D+ CD8+ T cell subsets, only transcripts that passed the microarray quality control were utilized. Analysis was performed on transcripts with the highest differential expression with respect to the KIR2D− population (Figure 5A). The analysis of DEGs showed that 356 genes in KIR2DL1/S1+ and 283 genes in KIR2DL2/L3/S2+ CD8+ T cells were specifically up-regulated compared to KIR2D− CD8+ T cells, whereas 118 genes were up-regulated in both KIR2D+ CD8+ T cell subsets. Besides, 449 genes in KIR2DL1/S1+ CD8+ T cells, 151 genes in KIR2DL2/L3/S2+ CD8+ T cells and 43 genes in both KIR2D+ CD8+ T cell subsets were down-regulatedcompared to KIR2D− CD8+ T cells (Figure 5B). Microarray analysis with specific data for these genes is shown in Appendix A. Microarray results were validated with qPCR of TYROBP, KIR3DL2, KLCR3, MEF2C, and FOSI2 genes (Figure 5C).

For further understanding of functional and metabolic pathways for DEGs involved in both KIR2D+ CD8+ T cell subsets, a functional enrichment analysis was performed with Metascape [21] (Figure 6). The analysis showed clear differences between KIR2DL1/S1+ and KIR2DL2/L3/S2+ CD8+ T cell subsets. The five most significant terms in KIR2DL1/S1+ CD8+ T cells were GO:0034655 Nucleobase-containing compound catabolic process, R-HSA-1280218 Adaptive immune system, R-HSA-69278 Cell cycle, mitotic, GO:0002366 Leukocyte activation involved in immune system, and GO:0051169 Nuclear transport; whereas for KIR2DL2/L3/S2+ CD8+ T cells the five most significant terms were hsa05330 Allograft rejection, hsa05322 Systemic lupus erythematosus, GO:0019083 Viral transcription, GO:0019221 Cytokine-mediated signaling pathway, and GO:0072657 Protein localization to membrane. Files S2 and S3 shows all up-regulated and down-regulated terms with their corresponding genes for KIR2DL1/S1+ and KIR2DL2/L3/S2+ CD8+ T cells subsets respectively.

Additionally, we were specifically interested in the expression of important moleculesimplicated in the effector function of cytotoxic T cell. As previously described, and in contrast to KIR2D−, KIR2DL1/S1+ and KIR2DL2/L3/S2+ CD8+ T cell subsets did not express CD28 (CD28 negative). Besides, all CD8+ T cell subsets barely expressed IL-2, IFNα, or IL-8, but they highly expressed IL-22, IL-32, IL18RAP, TGFβ, and IFNγ, although KIR2DL2/L3/S2+ CD8+ T cells showed higher expression of IFNγ (444 vs. 105 and 112, normalized intensity) and TGFβ (352 vs. 241 and 254, normalized intensity) than KIR2DL1/S1+ and KIR2D− CD8+ T cells. In contrast, KIR2DL1/S1+ CD8+ T cells showed lower expression of IL18RAP (also known as IL-1R5, 39 vs. 215 and 175, normalized intensity) than KIR2DL2/L3/S2+ and KIR2D− CD8+ T cells. Additionally, all CD8+ T cell subsets showed high expression of perforin and granzymes A, B, and H. Nonetheless, both KIR2D+ CD8+ T subsets expressed higher levels of granzymes than the KIR2D− subset. KIR2DL2/L3/S2+ CD8+ T cells showed the highest expression of perforin (Figure 5D).

### 2.6. KIR2D+ CD8+ Functional Pathways

For a more comprehensive analysis of molecular pathways involved in the immune response of KIR2DL1/S1+ and KIR2DL2/L3/S2+ CD8+ T cell subsets, the Protein-Protein Interaction (PPI) networks of specific up- and down-regulated genes in each subset were analyzed by means of theSTRING database [22]. To identify hub genes in the PPI networks, Degree and MCC algorithms of Cytohubba tool were used [23]. The results showed 34 up-regulated and 42 down-regulated genes in KIR2DL1/S1+ CD8+ T cells. In KIR2DL2/L3/S2+ CD8+ T cells 32 and 25 genes were identified as a hub in the up and down-regulated groups, respectively.

In KIR2DL1/S1+ CD8+ T cells, cluster analysis of the PPI network identified different modules with up-regulatedgenes that were involved in: (1) regulating protein expression by controlling the transcriptional status of cells (GO-0006351 Transcription, DNA template); (2) signaling pathways of inflammatory and immediate allergic reactions (leading to the release of potent inflammatory mediators, such as histamine, proteases, chemotactic factors or cytokines) or the phosphorylation of several proteins which regulate cell proliferation, development, and survival of cells (HSA-2454202 Fcε receptor -FCERI- signaling); and (3) maintenance of proper cell functioning by cleaving intracellular foreign or aberrant host proteins (HSA-983168 Antigen processing: Ubiquitination & Proteasome degradation). In contrast, other modules showed down-regulatedgenes that were involved in: (1) regulating different phases of glycolysis, one of the main energy systems of cellular metabolism (GO:0061621 canonical glycolysis); and (2) cell cycle, including structural genes involved in the mitotic and nuclear division and regulatory genes controlling different phases of the cell cycle (HSA-69278 Cell Cycle, Mitotic, HSA-69620 Cell Cycle Checkpoints and finally, GO:0000280 nuclear division) (see Figure 7 for a detailed list of genes).

In KIR2DL2/L3/S2+ CD8+ T cells cluster analysis of the PPI network identified different modules with up-regulatedgenes that were involved in: (1) signaling pathways of chemokine and cytokine receptors, promoting the release of effector proteins for tumor immune response, such as INFγ or perforin (GO:0019221 cytokine-mediated signaling pathway); and (2) signaling pathways constitutively activated in cancer (HSA-2219528 PI3K/AKT Signaling in Cancer). Modules with down-regulatedgenes in these cells were related to: (1) ribosomal subunit proteins and proteins that impair the proper translational and post-translational processes (GO:0006413 translational initiation and GO:0006614 SRP-dependent cotranslational protein targeting to membrane); (2) genes regulating intracellular K^+^ ion and its effect on cellular activity (HSA-1296072 Voltage gated Potassium channels); and (3) genes whose effects in cell signaling, cancer development or immune response are currently under examination (GO:0040029 regulation of gene expression, epigenetic) (see Figure 8 for a detailed list of genes).

## 3. Discussion

In line with previous results describing the expansion of KIR2DL1+ CD8+ T cells in nonmetastatic melanoma patients [14], the data in this manuscript show that HLA C2-ligands drove the expansion of KIR2DL1+ CD8+ T lymphocytes in patients with 3 different types of solid cancers (melanoma, ovarian carcinoma and bladder carcinoma). The immunosurveillance functions of KIR2DL1+ CD8+ T lymphocytes might have contributed to keeping patients alive during the follow-up period and suggest a protective role for these cells, since the expansion of KIR2DL1+ CD8+ T lymphocyteswas completely abrogated in patients who died. In contrast, and regardless of tumor type, HLA C1-ligands drove a dose-dependent expansion of KIR2DL2/S2+ CD8+ T cells in patients whose tumor growth could not be controlled and who therefore died during the follow-up period, which supports a suppressive/regulatory function for KIR2DL2/S2+ CD8+ T lymphocytes. These data suggest that the interaction of specific tumor-associated ligands [24] could induce the expansion of different KIR+ CD8+ T cell subsets that decisively contribute to the metastatic dispersion of the tumor and the outcome of the patient. Supporting this idea, our results show that patients carrying HLA C1-ligands showed shorter overall survival in the Kaplan-Maier analysis.

In an attempt to understand the mechanisms underlying the differential functioning of CD8+ T cells depending on the type of KIR receptor they express, comparative transcriptomic studies among KIR2D−, KIR2DL1/S1+, and KIR2DL2/L3/S2+ CD8+ T lymphocytes were carried out. The results of this study show that far from simply modulating T cell receptor (TCR) signaling [12,13], interactions of KIR receptors with their specific ligands have a profound differential impact on the repertoire of expressed genes, thus conditioning multiple signaling pathways, effector functions, the secretome, and ultimately affecting the tumor microenvironment.

KIR2DL1/S1+ CD8+ T cells show over-expression of components of the FcεRI and NCR3 signaling pathways, genes which play an important role in tumor killing by T and NK cells [25,26]. Sos1, GNG2, and Rac1 genes, in the Ras signaling pathway, are also over-expressed. Sos1 plays an important role in T cell differentiation, proliferation, and effector functions, and triggers the ERK pathway [27]. Rac1 activation triggers migration and proliferation of immune cells in response to chemokines [28]. In the same node, the serine/threonine kinase MAPK8, which is also involved in T cell signaling, was up-regulated as well [29]. Besides, two important transcription factors that can improve T and NK cell anti-tumor effector function were over-expressed: NFκβ1, which is involved in the development, differentiation and activation of innate and adaptive immune cells [30], and SIRT1, a NAD+ dependent protein deacetylase [31]. Components of the proteasome (PSMA1, PSMA8 and PSMA9), NEDD8 and other E3 ubiquitin proteasome ligands are also over-expressed. The proteasome is necessary for MHC antigen-presentation and regulates metabolism and differentiation of CD8+ T lymphocytes [32]. NEDD8 triggers downstream signals modulating antitumor responses of innate and T cell [33] by means of neddylation, an analogous process to ubiquitination. On the other hand, KIR2DL1/S1+ CD8+ subset exhibited down-regulatedgenes encoding proteins involved in cell cycle regulation (CDK1, CCNA2, CCNB1/2) [34] and cell division (KIF18B, KIF2C, CENPH, NCAPD2, NCAPH, and NCAPG2) suggesting a proliferative arrest after 19 days of intensive expansion. Supporting this idea, c-Myc, ENO1, PKM, TPI1 and GAPDH, genes with important roles in T cell development, differentiation, proliferation, and glycolysis [35], were also down-modulated.

On the whole, gene expression profile of KIR2DL1/S1+ CD8+ cells defines a T cell subset with high anti-tumor potential that remains in a resting state after the in vitro expansion. At this point, it is important to highlight that our results show that, in the appropriate conditions, both KIR2DL1/S1+ and KIR2DL2/L3/S2+ CD8+ T cells are capable of expanding at least as intensely as KIR2D negative CD8+ T cells, which is in contrast to what has been previously establishedin the literature, where KIR-expressing CD8+ T cells have been described as highly differentiated cells with low proliferative capacity [10].

KIR2DL2/L3/S2+ CD8+ T lymphocytes exhibited over-expression of genes that are not usually expressed in CD8+ T cells, i.e., EGFR, ITGB1, MAP2K1, and CD86, indicating that these cells are reprogrammed to the AKT/PI3K cancer pathway. This could be related to the down-regulation of DNMT3B, a DNA (cytosine-5)-methyltransferase 3B required for de novo DNA methylation. DNA methylation governs gene expression potential and hence cell identity. Besides, epigenetic marks can be transmitted to daughter cells during antigen-driven clonal expansion and thus enable the fast and efficient re-activation of genes coding for critical effector molecules in memory cells [36]. Consequently, its down-regulation could impair the correct functioning of KIR2DL2/L3/S2+ CD8+ T cells. Over-expression of TNFRSF1B, FoxO3, CD244 and TGFBR1 suggests an immune-suppressive role of these CD8+ T cells. TNFRSF1B controls the expression of the inhibitory receptor Tim3 [37] and promotes cell death [38]. FoxO could inhibit T cell effector function through inhibition of T-bet expression [39]; CD244 could be indicative of an exhausted phenotype and is related to cancer immune tolerance [40]; and finally TGFBR1 is the main receptor for the immunosuppressive cytokine TGFβ [41].

The up-regulation of CD44 observed in KIR2DL2/L3/S2+ CD8+ T cells will favor their interaction with VLA-4 integrin in order to facilitate their infiltration in the tumor [42] and their contact with dendritic and other immune cells [43]. This is particularly relevant since these cells also over-express CCR1 and CCR5, chemokine receptors for ligands secreted by tumor cells [40,44] which are shared with monocytes/macrophages, dendriticcells, and neutrophils. This could bring together T cells secreting notable amounts of IFNγ, TGBβ, IL32, and IL18RAP with myeloid/monocyte-derived cells in the tumor microenvironment. It has been described that low concentrations of IFNγ promote inflammation, but once a certain concentration is reached, its antiproliferative and proapoptotic effects become dominant by inducing NO production in macrophages and/or myeloid cells [45,46]. IL-32 promotes differentiation of mature monocyte-derived dendritic cells and macrophages, which can express immune-suppressive indoleamine 2,3-dioxygenase (IDO) and IL-10 [47]. Besides, the interaction between IL-18 and the up-regulated IL18R can limit Th17 cell differentiation and promote the development of Foxp3+ Treg effectors [48]. TGBβ, highly secreted by the KIR2DL2/L3/S2+ CD8+ T cells, which in turn also over-expressed the TGFBR1, could have further sustained tumor evasion by promoting an immune-suppressive tumor microenvironment [41]. This hypothesis is supported by our in vitro results, which show that both IFNγ and TGFβ strongly suppressed the proliferation of all CD8+ T cell subsets.

Down-regulation of genes of the voltage-gated potassium channel pathway (KCNQ1 and KCNC3) could also contribute to the immune-suppressive profile of KIR2DL2/L3/S2+ CD8+ T cells, by leading to cytoplasm K+ accumulation, which could disrupt their effector functions [49]. However, the biggest PPI down-regulated module corresponds to various 40S or 60S ribosomal proteins, which could impact protein translation (GO:0006413 and GO:0006614), which is essential for the correct functioning of these cells. Thus, down-expressed RPS3A can compromise TCR-mediated activation through NF-κβ signaling pathway, since RPS3A is a non-Rel component of the NF-κB complex that directly binds to the RelA subunit, favoring NF-κB DNA-binding and transcription of specific genes [50].

In summary, although KIR2DL2/L3/S2+ CD8+ T lymphocytes express high amounts of NK-cell-related cytotoxicity molecules (perforin, granzyme, etc.), significant changes in their expression profile suggest a negative impact on their own immunosurveillance effector functions and their possible involvement in the promotion of several other immune suppressive mechanisms.

In conclusion, and although a direct role of KIR+ CD8+ T cell in the containment or escape of cancer should be validated in future functional assays, our data show that KIR receptors expressed on CD8+ T lymphocyte control the expansion and function of these T cells by interacting with their specific ligands. In patients who survive long periods after the cytoreductive therapy, HLA C2-ligands favor the expansion of KIR2DL1+ CD8+ T cells, which show a gene expression signature related to efficient tumor immunosurveillance. In contrast, HLA C1-ligands guide the expansion of KIR2DL2/S2+ CD8+ T cells that are associated to shorter survivals and show transcriptomic profiles related to suppressive anti-tumor responses. These results could be relevant for the treatment of patients with different types of solid cancer, since the expansion of suppressive KIR2DL2/S2+ CD8+ T cells in patients with HLA C1-ligands could be reversed with anti-KIR therapies, and conversely, expansion in vitro of KIR2DL1/S1+ CD8+ T cells favored by IL-12 could be assayed to determine its effectiveness as an anti-tumor cell therapy.

## 4. Materials and Methods

### 4.1. Healthy Controls and Patient Characteristics

This prospective observational study included 42 healthy Caucasian volunteers (control group) and 249 consecutive Caucasian patients with melanoma (*n* = 80), bladder (*n* = 80), or ovarian (*n* = 89) tumors (Table 1). These series of controls and cancer patients coincide with the cohorts used in previous studies, whose aim were to assess the role of inhibitory and activating KIR receptors on NK cell education and tumor immune surveillance [18,20]. Peripheral blood samples anticoagulated with EDTA were obtained from controls and patients from the Clinic UniversityHospital Virgen de la Arrixaca and General University Hospital Santa Lucía (Murcia, Spain). Expression of inhibitory (KIR2DL1, KIR2DL2/S2, KIR2DL3, and KIR3DL1) and activating (KIR2DS1) KIR receptors on CD3+CD4+ and CD3+CD8+ T cells and CD3−CD56+ NK cells were analyzed in peripheral blood samples obtained at diagnosis, prior to the oncologic treatment. An Institutional review board (IRB-00005712) approved the study. Written informed consent was obtained from every patient and control in accordance with the Declaration of Helsinki.

### 4.2. Immunophenotyping of CD8+ Peripheral Blood T Cells at Diagnosis

The expression of KIR receptors (KIR2DL1, 2DS1, 2DL2/S2, 2DL3 and 3DL1) on CD3+CD8+ T cells and CD3–CD56+ NK cells in peripheral blood was evaluated as a percentage of positive cells using LSR-II and DIVA Software (BD, San Diego, CA, USA), as previously described [18,20]. LSR-II photomultiplier (PMT) voltages were adjusted daily using rainbow calibration particles (BioLegend, San Diego, CA, USA). Fluorescence compensation was finely adjusted using negative events for each fluorochrome as a reference. The staining protocol consisted of a 11-color/12-monoclonal antibody (mAb) panel: CD3 AmCyam (clone SK7, BD), CD4 PE-CF594 (RPA-T4, BD), CD8 APCCy7 (SK1, BD), CD16 PacBlue (3G8, BD), CD56 BV711 (NCAM16.2, BD), CD158a,h PECy7 (EB6, BD, recognizes both KIR2DL1 and 2DS1), CD158b1/b2,j PE-Cy5 (GL183, Beckman-Coulter, Brea, CA USA, recognizes KIR2DL2, 2DL3 and 2DS2), CD158a FITC (143211, R&D Systems, MN, USA; KIR2DL1), CD158b2 APC (180701, R&D Systems, KIR2DL3), CD158e APC (DX9, R&D Systems, KIR3DL1), CD226 PE (11A8, BioLegend), and NKG2A biotin (REA110, Miltenyi Biotech, BergischGladbach, Germany).

The gating strategy used to identify total lymphocytes, CD3+, CD4+, and CD8+ T lymphocytes, as well as CD16^−/+^CD56^++^ (CD56^bright^) and CD16^+^CD56^+^ (CD56^dim^) NK cells is shown in Appendix A.

### 4.3. HLA-A, -B and -C and KIR Genotyping

HLA-A, -B, -C and KIR genotyping was performed in DNA samples extracted by QIAmp DNA Blood Mini kit (QIAgen, GmbH, Hilden, Germany) using sequence-specific oligonucleotide PCR (PCR-SSO) and Luminex^®^ technology with Lifecodes HLA-SSO and KIR-SSO typing procedure (Immucor Transplant Diagnostic, Inc. Stamford, CT, USA), as previously described [18,20]. HLA-C genotyping allowed distinction between HLA-C^Asn80^ (group-C1) and HLA-C^Lys80^ (group-C2) alleles [51]. HLA-A and -B genotyping allowed us to distinguish alleles bearing the Bw4 motif according to the amino-acid sequences at positions 77–83 in the α1 domain of the HLA class-I heavy chain.

KIR genotyping identified iKIRs (KIR2DL1, KIR2DL2, KIR2DL3, KRIR2DL5, KIR3DL1, KIR3DL2, and KIR3DL3) and aKIRs (KIR2DS1, KIR2DS2, KIR2DS3, KIR2DS4, KIR2DS5, and KIR3DS1), as well as KIR2DL4 that exhibits both inhibitory and activating potential. The method used could not distinguish between KIR2DL5A (telomeric) and KIR2DL5B (centromeric) forms. Different allotypes of KIR2DS4 were detected, including the expressed allotype KIR2DS4 full exon-5 (KIR2DS4full) and the non-expressed KIR2DS4 deleted exon-5 (KIR2DS4del).

### 4.4. In Vitro Expansion and Isolation of KIR2D−, KIR2DL1/S1+, and KIR2DL2/L3/S2+ CD8+ T Cells

Peripheral blood mononuclear cells (PBMC) were isolated from healthy volunteers with different HLA-C and KIR genotypes (volunteer-1: C1C1 with KIR2DL1+S1+ KIR2DL2/S2+L3+; volunteer-2: C1C2 with KIR2DL1+S1- KIR2DL2/S2+ L3+; and volunteer-3: C2C2 with KIR2DL1+S1- KIR2DL2/S2-L3+) over Ficoll-Hypaque (Lymphoprep^TM^, StemCell Technologies, Grenoble, France) density gradients. For the expansion of KIR2D+ CD8+ T cells, 10 × 10^6^ PBMCs per flask, as responding cells (R), were co-cultured in a mixed lymphocyte reaction (MLR) with 5 × 10^6^ 25Gy-irradiated PBMCs from HLA-C C1C2 donor as stimulating cells (S) and 0.5 × 10^5^ 100Gy-irradiated JY cell line as feeder, as described by Levings et al. [52]. These assays were performed three times for each genotype. To evaluate the influence of different cytokines in the expansion of KIR2D+ CD8+ T cells, IL-10 (20 ng/mL), IL-12 (5 ng/mL), IL-15 (10 ng/mL), IFNγ (50ng/mL), or TGFβ (1 ng/mL) were added to the MLR culture.

T cell immunophenotype was evaluated pre-expansion and after 9, 13, and 19 days of culture through staining with anti-CD3 BV510, anti-CD56 BV421, anti-CD8 APC-Cy7, anti-CD4 AF700, anti-CD158a/h PE, and anti-CD158b/j PE-Cy5 and then analyzed on a LSR-II flow cytometry (BD). KIR2D− and KIR2D+ CD8+ T cell proliferation was estimated as a fold change compared to day 0 through cell counting with a Neubauer chamber. Perforin expression was analyzed by intracellular staining with anti-perforin-FITC. On day 19, expanded cells were harvested and stained as above to isolate KIR2D− (KIR2DL1/S1 and KIR2DL2/L3/S2 negative), KIR2DL1/S1+ and KIR2DL2/L3/S2+ CD8+ T lymphocytes through FACS-sorting using MoFlow (Beckman Coulter, Brea, CA USA) and achieving purities higher than 98%.

### 4.5. RNA Extraction from KIR2D- and KIR2D+ CD8+ T Cells

Total RNA was extracted using the RNeasy mini kit (Qiagen) according to the manufacturer’s instructions. RNA pools were prepared by mixing equal amounts of RNA from sorted-cells of the three cell expansion experiments. Identical aliquots of each pool were used for microarray analysis after treatment with DNase-I Amplification Grade (Invitrogen, Thermo Fisher Scientific, Waltham, MA, USA). Other aliquots of the same pools were reverse-transcribed using the Superscript III kit (Invitrogen) and then used to validate the microarray data with quantitative polymerase chain reaction (qPCR).

### 4.6. T7 Amplification and Microarray Performance

Whole genome expression analysis was performed on CodeLink Human Bioarrays 36K genes obtained from Applied Microarrays (Tempe, AZ, USA) according to manufacturer’s protocols. These microarrays comprised 54,841 single-stranded 30-mer oligonucleotide probes for human genes, where approximately 57,000 transcripts were identified.

Briefly, 1 mg of high-quality total RNA was reverse transcribed using T7-oligo-dT primer and double-stranded complementary DNA (cDNA). After reverse transcription cRNAs were labeled with Cy5 Streptavidine (Amersham Biosciences, Ume, Sweden). Hybridization of whole genome human genes included in CodeLink Bioarrays (Applied Microarrays) was performed overnight at 37 ºC in an incubator shaker (Innova 4080, New Brunswick, NJ, USA). One hybridization reaction per condition was performed. Microrrays were read with a GenePix 4000B laser scanner (Axon Instruments, Union City, CA, USA), and then quantified, and normalized using CodeLink Software 5.0 (Applied Microarrays).

### 4.7. Array Data Analysis

Normalized data from the CodeLink software package were analyzed and only the genes that passed the CodeLink Bioarray quality controls were selected. Gene expression for microarray meta-analysis was computed pairing each KIR2D+ subset with the KIR2D− group. Significance was assessed using the following criterion: at least a 2-fold change in expression between groups. Groups of specific differentially expressed genes (DEGs) were identified by using jvenn plug-in (jvenn.toulouse.inra.fr) [53] and then summarized in a Venn diagram.

The transcripts selected by using the above-mentioned procedures were further analyzed using Metascape web-based portal [21]. Metascape is a powerful web-based tool for gene annotation and gene set enrichment analysis which incorporates different updated ontology databases such as KEGG Pathway, GO Biological Processes, Reactome Gene Sets, Canonical Pathways and CORUM. To perform the analysis, all genes in the genome were used as the enrichment background. Terms with a *p*-value < 0.01, a minimum count of 3, and an enrichment factor >1.5 were collected and grouped into clusters based on their membership similarities.

### 4.8. Real-Time Quantitative Reverse Transcription-PCR

Microarray data were validated by quantitative real-time PCR. Transcripts were quantified by using Applied Biosystems predesigned TaqMan Gene Expression assay (AB assays:Hs01050117for FOSL2, Hs01652462 for KLRC3, Hs00601497 for KIR3DL2, Hs00430498for LAIR2, Hs00182426 for TYROBP and Hs00231149 for MEF2C) on a ABI PRISM 7000 Sequence Detection System (Applied Biosystems), according to the manufacturer’s instructions. Relative expression of the target transcripts in KIR2DL1+ and KIR2DL2/L3/S2+ CD8+ T cells were calculated with the cycling threshold method as 2^−ΔΔCt^ relative to the expression of KIR2D− CD8+ T cells. Differential gene expression was considered significant when *p* < 0.05 in three independent cell preparations. Deviations of mRNA levels of each experiment (three individual cell samples run in duplicate) were first normalized to the expression of the housekeeping gene GAPDH RNA in that sample.

### 4.9. Protein-Protein Interaction Network Analysis

In this study, Protein-protein Interaction (PPI) networks of DEGs from KIR2DL1/S1+ and KIR2DL2/L3/S2+ CD8+ T lymphocytes were constructed using the online database Search Tool for the Retrieval of Interacting Genes (STRING; http://string-db.org) (version 11.0) [22]. An interaction with a combined score >0.4 was considered statistically significant. Cytoscape (version 3.8.0) is an open source bioinformatics software platform for visualizing molecular interaction networks [54]. Cytohubba plug-in was used to identify the most significant nodes in the protein network through Degree and Maximal Clique Centrality (MCC) algorithms [23]. Finally, these nodes were represented again in the STRING database, where the functional and enrichment analyses of these pathways were performed through its analysis algorithms.

### 4.10. Statistical Analysis

All data were collected in a database (Excel2003; Microsoft Corporation, Seattle, WA, USA) and analyzed with SPSS-15.0 (SPSS Inc., Chicago, IL, USA). ANOVA and post hoc tests were used to analyze continuous variables. Data were expressed as mean ± SEM. Kaplan-Meier estimator and Log-rank tests were used to analyze patient overall survival (OS). Time to death was estimated in months since the date of diagnosis. *p* Values < 0.05 were considered significant. Statistics associated with microarray analysis are described above.

## 5. Conclusions

CD8+ T cells expressing KIR receptor accumulate in the peripheral blood of oncologic patients. Interestingly, interactions of KIRs with their specific ligands have a profound impact on CD8+ T cell expression profiles, involving multiple signaling pathways, effector functions, the secretome, and consequently, the cellular microenvironment, which could impact their cancer immunosurveillance capacities. Thus, in patients who survived long periods after the cytoreductive therapy, HLA C2-ligands favored the expansion of KIR2DL1+ CD8+ T cells, which showed a gene expression signature related to efficient tumor immunosurveillance. In contrast, HLA C1-ligands guided the expansion of KIR2DL2/S2+ CD8+ T cells that were associated to shorter survivals and show transcriptomic profiles related to suppressive anti-tumor responses. These results could be the basis for the discovery of new therapeutic targets so that the outcome of patients with cancer can be improved.

## Figures and Tables

**Figure 1 cancers-12-02991-f001:**
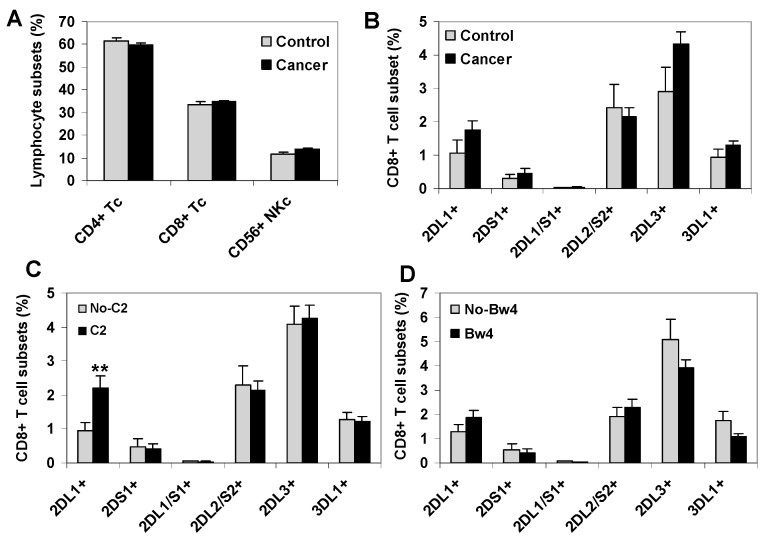
Peripheral blood KIR+ CD8+ T cell repertoire in healthy controls and solid cancer patients. (**A**) Frequency of mayor CD4+ and CD8+ T cell subsets and total CD56+CD3− NK cells in peripheral blood of healthy controls (*n* = 42) and cancer patients (80 melanomas, 80 bladder cancers and 89 ovarian cancers). (**B**) Frequency of KIR+ CD8+ T cell subsets in controls and cancer patients. (**C**,**D**) Frequency of KIR+ CD8+ T cell subsets in cancer patients according to the presence of the HLA-C2 or Bw4 ligands, respectively. ** *p* < 0.01, comparing KIR2DL1+ CD8+ T cells in HLA-C2 positive (C1C2 or C2C2) and negative (C1C1) cancer patients. Data represent frequency in total lymphocytes of different T and NK cells subsets.

**Figure 2 cancers-12-02991-f002:**
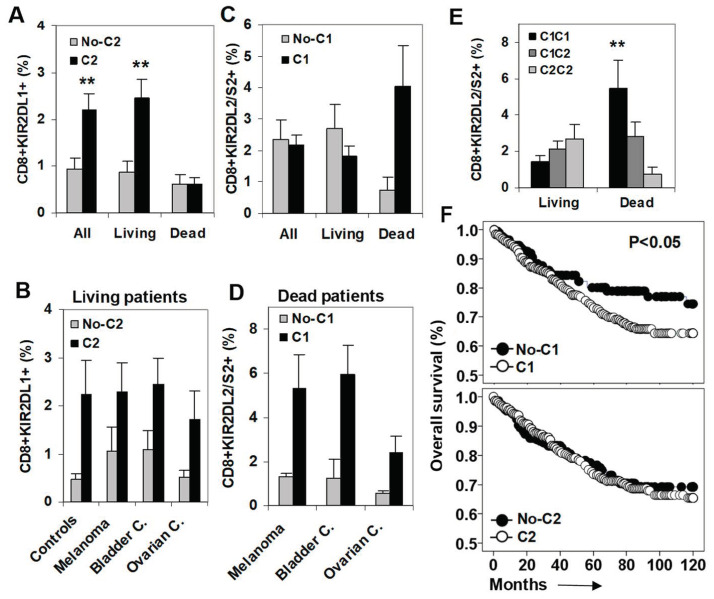
Impact of C2- and C1-ligands on the frequency of KIR2DL1+ and KIR2DL2/S2+ CD8+ T cell subsets and on patient survival. (**A**) Frequency of KIR2DL1+ CD8+ T cells (% of total lymphocytes) in all patients (*n* = 249), patients who survived (Living, *n* = 208), and patients who died during the follow-up (Dead, *n* = 41) according to the presence of the specific C2-ligand. ** *p* < 0.01. (**B**) Frequency of KIR2DL1+ CD8+ T cells in healthy controls and in living melanoma, bladder, and ovarian cancer patients. (**C**) Frequency of KIR2DL2/S2+ CD8+ T cells in all, living, and dead patients according to the presence of their specific C1-ligand. (**D**) Frequency of KIR2DL2/S2+ CD8+ T cells in dead melanoma, bladder, and ovarian cancer patients. (**E**) Frequency of KIR2DL2/S2+ CD8+ T cells in living and dead cancer patients according to the dose of its specific C1-ligands. (**F**) Kaplan-Meier and Log-rank tests for overall survival (OS) of solid cancer patients (*n* = 248) according to the presence of C1- and C2-ligands.

**Figure 3 cancers-12-02991-f003:**
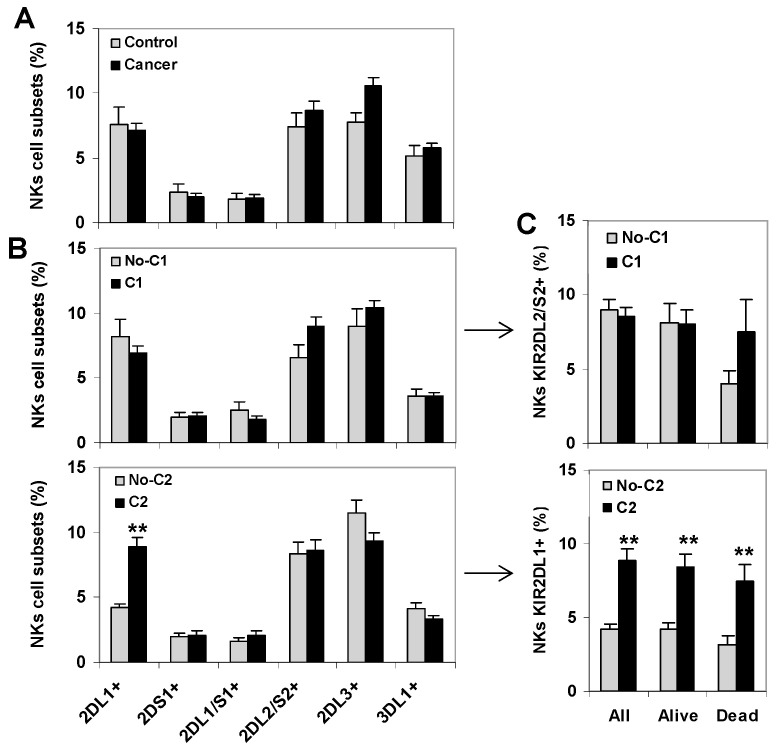
KIR+ NK cell repertoire in healthy controls and cancer patients. (**A**) Frequency of KIR+ NK cells in peripheral blood of healthy controls (*n* = 42) and cancer patients (80 melanomas, 80 bladder cancers and 89 ovarian cancers). (**B**) Frequency of KIR+ NK cell subsets in cancer patients according to the presence of the HLA C1- and C2-ligands, respectively. ** *p* < 0.01, comparing KIR2DL1+ NK cells in HLA-C2 positive (C1C2 or C2C2) and negative (C1C1) cancer patients. (**C**) Frequency of KIR2DL2/S2+ and KIR2DL1+ NK cells in all patients (*n* = 249), patients who survived.

**Figure 4 cancers-12-02991-f004:**
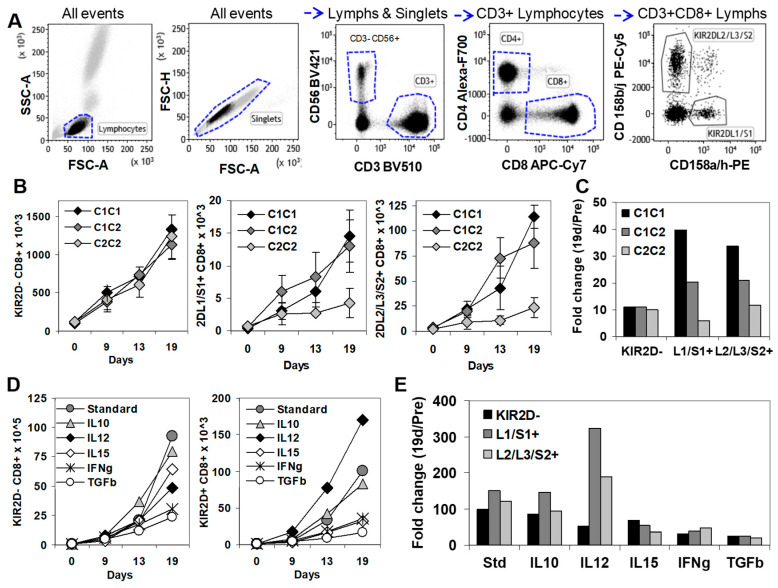
In vitro expansion of KIR2D− and KIR2D+ CD8+ T cells. (**A**) Immunophenotype analysis of KIR2D− and KIR2D+ CD8+T cells. Left to right, FSC/SSC dot plot showing all events to select lymphocytes, FSC-A/FSC-H dot plot showing all events to select singlets, CD3/CD56 dot plot showing singlet/lymphocytes to select CD3+ T cells and CD3−CD56+ NK cells, CD4/CD8 dot plot showing CD3+ lymphocytes to select CD4+ and CD8+ T cell subsets, and CD158ah/CD158bj dot plot showing CD8+ lymphocytes to select CD158ah (KIR2DL1/S1+) and CD158bj (KIR2DL2/L3/S2+) CD8+ T cells. (**B**) In vitro expansion of KIR2D− and KIR2D+ CD8+ T cell subsets from peripheral blood mononuclear cells (PBMC) of C1C1, C1C2, and C2C2 healthy donors in the presence of IL-2 (40 µg/mL), 0.5 × 10^6^ 100 Gy irradiated JY cell line, and 5 × 10^6^ 25 Gy irradiated PBMC from the same C1C2 healthy donor, at days 0, 9, 13, and 19 (standard conditions). These experiments were repeated 3 times. (**C**) Fold increase at day 19 of CD8+ T cell subsets in C1C1, C1C2, and C2C2 donors. (**D**) In vitro expansion of KIR2DL1/S1+ and KIR2DL2/L3/S2+ CD8+T cell subsets from PBMC of a C1C2 donor in the same conditions as in B, but IL-10 (20 ng/mL), IL-12 (5 ng/mL), IL-15 (10 ng/mL), IFNγ (50 ng/mL), or TGFβ (1 ng/mL) were added at day 0. (**E**) Fold change at day 19 of CD8+ T cell subsets after the addition of different cytokines to the standard expansion protocol.

**Figure 5 cancers-12-02991-f005:**
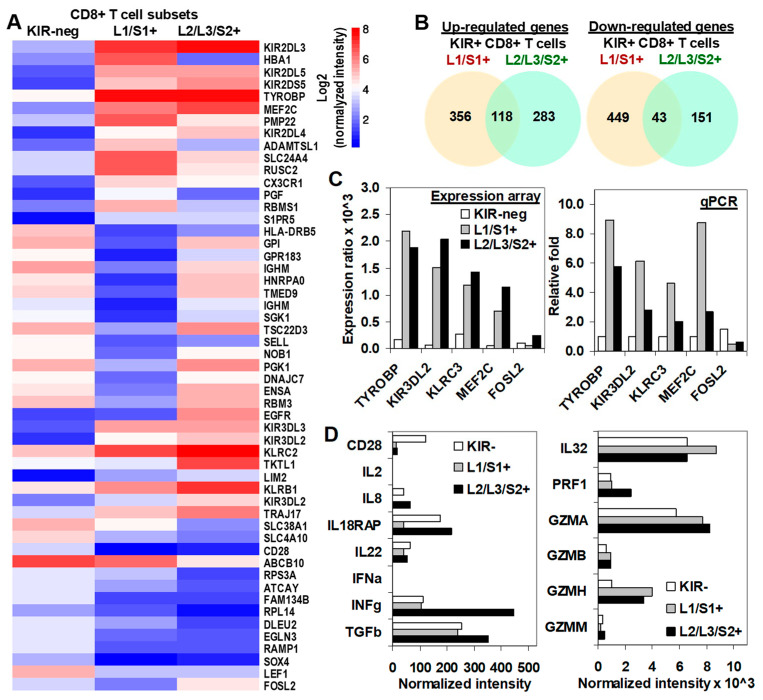
Transcriptional profiling of KIR2D− and KIR2D+ CD8+ T cells. (**A**) Differential gene expression profile (CodeLink Genome Array) of KIR− (column 1), KIR2DL1/S1+ (column 2), and KIR2DL2/L3/S2+ (column 3) CD8+ T cell subsets. The figure shows the top 40 most significantly modulated genes (designed with http://www.ehbio.com/ImageGP/). (**B**) Total number of up- and down-regulated genes in each KIR2D+ CD8+ T cell subset (analysis performed with jvenn software). (**C**) Microarray results were confirmed with real time quantitative RT-PCR (qPCR) analysis. Expression ratio (upper plot) and relative fold changes (lower plot) are shown for microarray and qPCR assays, respectively. Differential gene expression was considered significant with *p* < 0.05 in three independent cells preparations. Mean fold-changes in gene transcript expression levels between KIR2D− and KIR2D+ were evaluated with 2^ΔΔCt^. (**D**) Shows the normalized intensity of important molecules in the cytotoxic T cell biology and interleukin or interleukin receptors.

**Figure 6 cancers-12-02991-f006:**
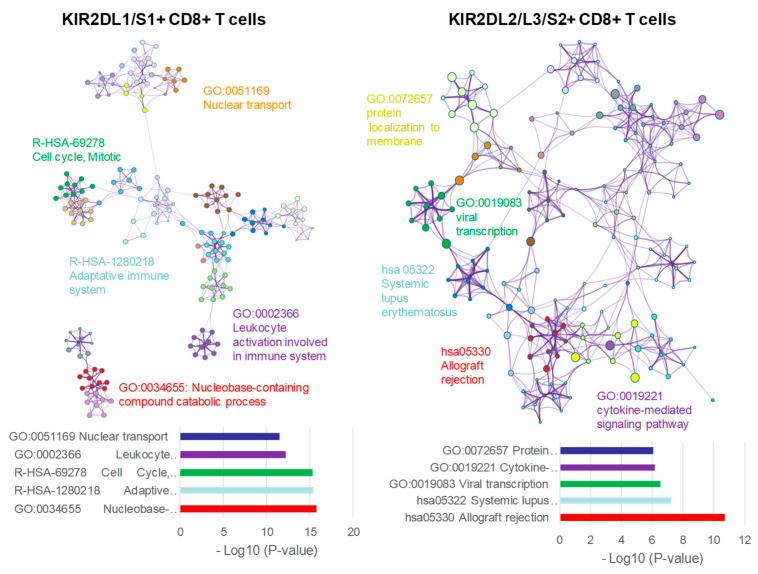
Functional enrichment analysis with Metascape of KIR2DL1/S1+ and KIR2DL2/L3/S2+ CD8+ T cell populations.Network of enriched terms colored by cluster ID, where nodes that share the same cluster ID are typically close to each other. On the left (KIR2DL1/S1+ CD8+ T cells) and right (KIR2DL2/L3/S2+ CD8+ T cells) DEGs where analyzed and the five top clusters with their representative enriched terms are shown.

**Figure 7 cancers-12-02991-f007:**
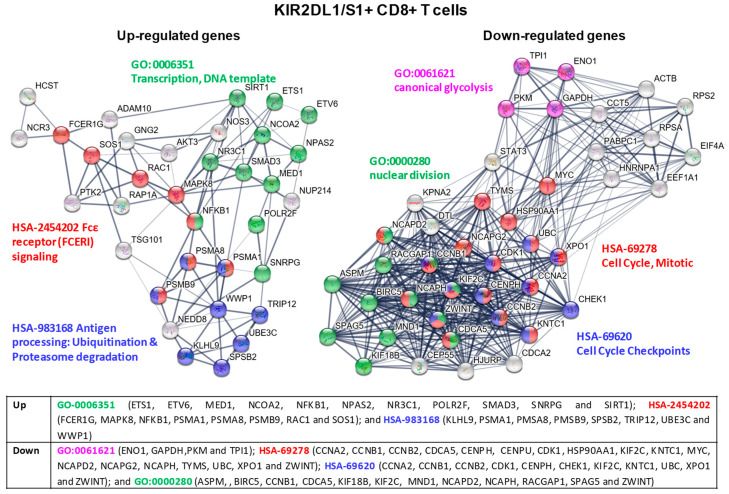
KIR2DL1/S1+ CD8+ T cell cluster analysis of the protein-protein interaction (PPI) network with STRING database. Hub nodes from differentially expressed genes were identified using Cytoscape software and then represented using STRING to perform the functional and enrichment analysis of these pathways. The PPI network shows up-expressed genes for modules GO-0006351 Transcription, DNA template (green), HSA-2454202 Fcε receptor (FCERI) signaling (red), HSA-983168 Antigen processing: Ubiquitination& Proteasome degradation (blue); and down-expressed genes for modules GO-0061621 canonical glycolysis (purple), HSA-69278 Cell Cycle, Mitotic (red), HSA-69620 Cell Cycle Checkpoints (blue), and GO:0000280 nuclear division (green). Briefly, the parameters used in STRING were as follows: text mining, experiments, databases, co-expression, neighborhood, gene fusion and co-occurrence were used as active interaction sources. The confidence threshold was set at 0.4. The disconnected transcripts were eliminated.

**Figure 8 cancers-12-02991-f008:**
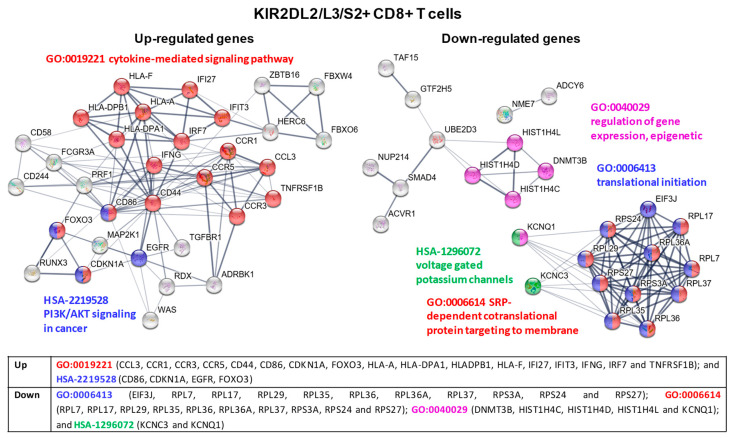
KIR2DL2/L3/S2+ CD8+ T cell cluster analysis of the protein-protein interaction (PPI) network with STRING database. Hub nodes from differentially expressed genes were identified using Cytoscape software and then represented using STRING to perform the functional and enrichment analysis of these pathways. The PPI network shows up-expressed genes for modules GO-0019221 cytokine-mediated signaling pathway (red) and HSA-2219528 PI3K/AKT Signaling in Cancer (blue); and down-expressed genes for modules GO:0006413 translational initiation (blue), GO-0006614 SRP-dependent cotranslational protein targeting to membrane (red), GO:0040029 regulation of gene expression, epigenetic (purple), and HSA-1296072 voltage gated potassium channels (green). Briefly, the parameters used in STRING were as follows: text mining, experiments, databases, co-expression, neighborhood, gene fusion and co-occurrence were used as active interaction sources. The confidence threshold was set at 0.4. The disconnected transcripts were eliminated.

**Table 1 cancers-12-02991-t001:** Demographic and clinical characteristics of patients and controls.

	Cancer Patients (*n* = 249)		
	Melanoma (*n* = 80)	Bladder (*n* = 80)	Ovarian (*n* = 89)
Sex (M/F, %M) ^1^	43/37 (53.7%)	68/12 (85%)	0/89 (0%)
Age (mean ± SD)	60.6 ± 15.4	71.8 ± 10.2	58.7 ± 10.7
Months since diagnosis (Mean ± SD)	4.6 ± 14.5	14.9 ± 32.8	7.0 ± 19.6
Histology (*n*) ^2^	35/45	36/44	69/20
Staging (*n*) ^3^	51/29	2/78	26/63
Treatment ^4^	10/9/61	51/20/9	32/49/8
Progression (yes/no, %)	27/80 (33.7%)	24/80 (30.0%)	37/89 (41.6%)
Death (yes/no, %)	7/73 (8.7%)	12/80 (15.0%)	20/89 (22.5%)

^1^ Control group: 42 healthy controls (59 ± 13.1 years, 37% males). ^2^ For melanoma (nodular/others), bladder (muscle invasive/superficial), ovarian (high-degree serous, undifferentiated and carcinomas/others). ^3^ For melanoma (Breslow’s depth <4 mm/>4 mm), bladder cancer (grade I or II/III), and ovarian carcinoma (grade I or II/III or IV). ^4^ For melanoma (interferon-alpha/vemurafenib/others), bladder cancer (BCG/cystectomy with or without chemotherapy/no treatment), and ovarian carcinoma (surgery + chemotherapy/neoadjuvant chemotherapy + surgery/palliative).

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
