# Peer review of "KIR+ CD8+ T Lymphocytes in Cancer Immunosurveillance and Patient Survival: Gene Expression Profiling"

_cancers, 2020, doi:10.3390/cancers12102991_

Round 1
Reviewer 1 Report
The authors have an impressive dataset of 249 cancer patients consisting of 80 melanoma, 80 bladder and 89 ovarian cancer patients. Overall, I am impressed with the analysis and results, although I feel the analysis and results could be tightened up a bit in order to improve the overall message of the paper prior to publication.
Introduction.
The authors mention the secretome and the importance of this in tumour immunology – are the authors aware of recent breakthroughs in NK cell receptor recognition of the tumour secretome e.g. NKp44 (Barrow et al. 2018) and NKG2D (Deng et al) and MICA (Younghoon Kim et al. Front Immunol, 2020) and how this may influence signalling from NK cells and CTL that express such receptors? These types of examples should be highlighted in the Introduction in terms of the tumor microenvironment and how it influences the activation and possible expansion of NK cells and CTL to give a broader perspective/readership on NK cell receptors signalling that e.g. KIR signalling may regulate as inhibitory receptors.
Results:
What about KIR+ NK cell populations? Surely KIR+ NK cell populations were analysed too? The results should really be included and discussed since the effect on overall survival may also be due to KIR+ NK cells not just KIR+ CTL.
Fgiure 1A: What are ‘CD4+ Tc’? The gating in Supplementary Fig 2 isn’t self-explanatory – the reader should be able to understand the gating without referring too much to the accompanying legend. Please indicate which plot leads to another with arrows from the parental gate and enhance this figure to show all KIR+ populations including the ‘CD4+ Tc’ for each cancer. CD4+ T cells have also been shown to express KIR (Van Bergen, J et al. J Immunol, 2004, 2009) and it is important to know if each cancer is similar or different when it comes to KIR expression on different NK and T cell populations/subsets that may reveal new and exciting insights.
Did the NKT cells ever express KIR? I expect not and so this would be a nice control.
Figure 2B and 2D: results from all three “solid tumours” are represented but these are quite different cancers and the breakdown for each cancer type should be presented in e.g. supplemental data if possible and discussed even if not significant.
Figure 3A – please make gating more clear – please indicate which plot leads to another with arrows from the parental gate.
Supplementary Figure S1. This is a beautiful analysis/figure and shouldn’t be hidden in the supplementary data – please include in the main manuscript.
Sections 2.4. Transcriptional profiling in KIR2D+ CD8+ T cells: Functional analysis and 2.5. KIR2D+ CD8+ functional pathways read a bit like a catalogue. Gene lists aren’t the most exciting so can these sections be enhanced by highlighting the relevance to tumor immunity?
Are any of these CD8+ T cell pathways expressed and have some effect on patient survival in the TCGA database for the cancers analysed? You could use GEPIA (http://gepia.cancer-pku.cn/) or other databases to analyse gene signatures representing these pathways in the cancers that might be associated with survival or positively correlated with T cell responses. It would be nice to see some type of analysis like this since it would validate your findings and show the significance of all these pathways for the cancers analysed and would enhance your paper.
Minor:
Line 113, separate words in ‘periodwith’
Line 116, shorter not sorter.
Author Response
To Reviewer-1
Comments and Suggestions for Authors
The authors have an impressive dataset of 249 cancer patients consisting of 80 melanoma, 80 bladder and 89 ovarian cancer patients. Overall, I am impressed with the analysis and results, although I feel the analysis and results could be tightened up a bit in order to improve the overall message of the paper prior to publication.
Introduction.
The authors mention the secretome and the importance of this in tumour immunology – are the authors aware of recent breakthroughs in NK cell receptor recognition of the tumour secretome e.g. NKp44 (Barrow et al. 2018) and NKG2D (Deng et al) and MICA (Younghoon Kim et al. Front Immunol, 2020) and how this may influence signalling from NK cells and CTL that express such receptors? These types of examples should be highlighted in the Introduction in terms of the tumor microenvironment and how it influences the activation and possible expansion of NK cells and CTL to give a broader perspective/readership on NK cell receptors signalling that e.g. KIR signalling may regulate as inhibitory receptors.
- We have reviewed and included these references in the introduction as suggested. Likewise, they have been related to the responses induced by KIR receptors. (lines 61-72)
Results:
What about KIR+ NK cell populations? Surely KIR+ NK cell populations were analysed too? The results should really be included and discussed since the effect on overall survival may also be due to KIR+ NK cells not just KIR+ CTL.
-Although the role of NK cells in the survival of patients of this series was described in previous manuscripts by our group, we have analyzed NK cell KIR repertoire with a similar strategy to the one used to analyze CD8+ T lymphocytes. We have included a new figure (Fig. 3) and a new section in the results (2.3) to describe results for NK cells. In summary, a clear impact on patient survival was not observed in relation to the frequency of KIR2DL1+ or KIR2DL2/S2+ NK cells. (lines 173-184)
Fgiure 1A: What are ‘CD4+ Tc’? The gating in Supplementary Fig 2 isn’t self-explanatory – the reader should be able to understand the gating without referring too much to the accompanying legend. Please indicate which plot leads to another with arrows from the parental gate and enhance this figure to show all KIR+ populations including the ‘CD4+ Tc’ for each cancer. CD4+ T cells have also been shown to express KIR (Van Bergen, J et al. J Immunol, 2004, 2009) and it is important to know if each cancer is similar or different when it comes to KIR expression on different NK and T cell populations/subsets that may reveal new and exciting insights.
-We have reorganized supplementary figure-1 (now supplementary Fig-2) in order to make it more understandable. Besides, we have rewritten the legend to make it clearer. Yes, we are fully aware that KIR receptors can be expressed by CD4+ T lymphocytes. However, both in cancer patients and in controls, the expression of KIR receptors in CD4+ T lymphocytes is marginal and with high variability among patients. For that reason, we did not monitor the expression of KIR in CD4+ T cells and unfortunately we have no data to offer. (line 429)
Did the NKT cells ever express KIR? I expect not and so this would be a nice control.
-We are sorry, but we have not studied NKT cells. Nonetheless, the KIR receptors analyzed by flow cytometry in this manuscript showed a very clear expression, which allowed an easy discrimination between positive and negative cells. Negative cells in each sample were used as internal controls to set positivity-markers for each KIR subset.
Figure 2B and 2D: results from all three “solid tumours” are represented but these are quite different cancers and the breakdown for each cancer type should be presented in e.g. supplemental data if possible and discussed even if not significant.
-We have added a supplementary figure-1 to describe data of KIR+ CD8+ T cell subsets for each type of cancer: melanoma, bladder and ovarian cancer. This figure is now cited in the manuscript. (lines 147-148)
Figure 3A – please make gating more clear – please indicate which plot leads to another with arrows from the parental gate.
-We have modified the figure to indicate de flux of gating. Legend has been adapted. (line 212)
Supplementary Figure S1. This is a beautiful analysis/figure and shouldn’t be hidden in the supplementary data – please include in the main manuscript.
-Of course.
Sections 2.4. Transcriptional profiling in KIR2D+ CD8+ T cells: Functional analysis and 2.5. KIR2D+ CD8+ functional pathways read a bit like a catalogue. Gene lists aren’t the most exciting so can these sections be enhanced by highlighting the relevance to tumor immunity.
-We have rewritten 2.4 and 2.5 sections and modified the respective figures to include the list of genes in the figures, thus avoiding its enumeration in the text. The text is now aimed more at describing the functions of each gene module. (lines 162-175) and (lines 178-189)
Are any of these CD8+ T cell pathways expressed and have some effect on patient survival in the TCGA database for the cancers analysed? You could use GEPIA (http://gepia.cancer-pku.cn/) or other databases to analyse gene signatures representing these pathways in the cancers that might be associated with survival or positively correlated with T cell responses. It would be nice to see some type of analysis like this since it would validate your findings and show the significance of all these pathways for the cancers analysed and would enhance your paper.
-We would like to thank the reviewer for such an interesting recommendation. We indeed have tried to validate our results with the GEPIA tool. Although it has been possible to select gene expression profiles (and include the up- and down- regulated genes in our KIR+ CD8+ T cells) for different types of T cells (effector or regulatory) and explore their impact in melanoma, ovarian and bladder cancers, GEPIA results, however, do not clearly validate our results. In this sense, it is important to take into account that the results of gene expression described in our manuscript have been obtained with a minor subset of peripheral blood CD8+ T cell that required in vitro expansion for 19 days and FACS-sorting. Therefore, we sincerely believe that the impact of the gene expression profiles described for KIR-expressing CD8+ T cells on patient survival cannot be compared with that of total effector or regulatory T cells. For this reason, we believe that far from helping to validate our results and favor their understanding, these results could cause confusion and we have therefore decided not to include them in the manuscript.
Please, have a look at the supplementary data attached “AT THE BOTTON of this point-by-point response” to see how the results seem nearly the same regardless of whether we select up- or down-regulated genes in KIR2DL1+ or KIR2DL2+ CD8+ T cells for melanoma, ovarian or bladder cancers.
Please, have a look at the supplementary data attached “AT THE BOTTON of this point-by-point response” to see how the results seem nearly the same regardless of whether we select up- or down-regulated genes in KIR2DL1+ or KIR2DL2+ CD8+ T cells for melanoma, ovarian or bladder cancers.
Minor:
Line 113, separate words in ‘periodwith’ Yes, thanks.
Line 116, shorter not sorter. Yes, thanks.
Reviewer 2 Report
Gimeno and colleagues present a well written research article summarising their findings on KIR+ CD8+ T cells in a range of solid cancer patients and extend these findings with in vitro approaches and extensive transcriptional analyses of cultured cells.
KIRD2L2+ and KIRD2L1/S1+ CD8+ T cell subtypes in the peripheral blood of solid cancer patients (melanoma, bladder, ovary) are investigated and frequencies over the course of disease, correlation with disease progression, and overall survival are reported. Specifically, the authors show a loss of KIRD2L1+ CD8+ T cells in the blood of HLA-C2+ patients that have succumbed to disease. Moreover, an increase in KIRD2L2+/S2+ CD8+ T cells in HLA-C1+ patients that had succumbed to cancer is shown, with HLA-C1+ individuals exhibiting a reduction in overall cancer survival rates.
These patient data are complemented by in vitro approaches investigating proliferative capacity and transcriptional profiles of KIR2D+/- CD8+ T cell subsets of healthy donors. The authors demonstrating that KIR2D+ CD8+ T cells preferentially expand in the presence of HLA-C1 and that this expansion is further enhanced by the presence of IL-12. Finally, the authors perform transcriptional profiling of in vitro expanded KIR2D+/- CD8 T cell subsets. Based on the analysis of differentially expressed genes and protein-protein interaction networks as well as the patient data, the authors conclude that KIRD2L1+ CD8+ T cells exhibit a profile of “efficient tumor immunosurveillance” while KIRD2L2+/S2+ CD8+ T cells present a profile consistent with an immune-suppressive phenotype.
Overall all, this is an interesting study which provides new insights on KIR+ CD8+ T cells in the context of solid cancers and the possible differential roles KIR+ CD8+ T cell subsets in the immune response to cancers. Although transcriptional analysis is limited by being performed on in vitro expanded KIR2D+/- CD8+ populations of healthy donors rather than directly on KIR2D+/- cells isolated form cancer patients, the extensive transcriptional characterisation, functional pathway analysis and in depth discussion provide novel and relevant insights on KIR+ CD8 T cell biology.
The authors may want to consider addressing the following points to further enhance the quality of the manuscript:
- In the introduction the authors state the interactions KIR subtypes KIRD2L1, D2L2/S2/L3 and 3DL1 are the best described. Further details on the current understanding of the role of these receptors in T or even NK cell biology in cancer settings would help put data presented in this manuscript into additional context.
- The subheading of the first results section states C2-ligand was associated with expansions of KIR2DL1+ CD8+ T cells in surviving patients. However, the data presented in Fig 2B shows that 2DL1+ CD8+ T cells are also expanded in healthy controls indicating that rather than the receptor upregulation correlating with protection, it is perhaps the loss of this receptor that correlates with poor survival? Did these patients have higher 2DL1+ CD8+ T cell frequencies at diagnosis? If the data is available it would be interesting to add.
- Do KIR2DL1 and KIRD2L2+/S2+ NK cells show similar patterns of loss and expansion in non-survivors as CD8+ T cells? And if so, could the overall survival be dependent on those subsets rather than CD8 T cells?
Minor comments
- Line 116-117 should read “….. and shorter patient survival”
- Line 130, the data referred to here is in Figure 2F
- Figure 3A, third panel should read CD3- CD56+ cells
Finally some more general thoughts that may be of interest to address in the future or include in the discussion
- Can KIR2D+ CD8 T cells be detected in the tumour microenvironment or only in the peripheral blood?
- Functional assays on the ability of the 2DL1+ CD8+ T cells to either directly affect cancer cells or to support the action of cytotoxic, tumour-specific CD8 T cells or NK cells might provide further insight. Similarly, the effects of KIRD2L2+/S2+ CD8+ T cells on tumour-specific populations could be tested in in vitro co-cultures.
Author Response
To Reviewer-2
Comments and Suggestions for Authors
Gimeno and colleagues present a well written research article summarising their findings on KIR+ CD8+ T cells in a range of solid cancer patients and extend these findings with in vitro approaches and extensive transcriptional analyses of cultured cells.
KIRD2L2+ and KIRD2L1/S1+ CD8+ T cell subtypes in the peripheral blood of solid cancer patients (melanoma, bladder, ovary) are investigated and frequencies over the course of disease, correlation with disease progression, and overall survival are reported. Specifically, the authors show a loss of KIRD2L1+ CD8+ T cells in the blood of HLA-C2+ patients that have succumbed to disease. Moreover, an increase in KIRD2L2+/S2+ CD8+ T cells in HLA-C1+ patients that had succumbed to cancer is shown, with HLA-C1+ individuals exhibiting a reduction in overall cancer survival rates.
These patient data are complemented by in vitro approaches investigating proliferative capacity and transcriptional profiles of KIR2D+/- CD8+ T cell subsets of healthy donors. The authors demonstrating that KIR2D+ CD8+ T cells preferentially expand in the presence of HLA-C1 and that this expansion is further enhanced by the presence of IL-12. Finally, the authors perform transcriptional profiling of in vitro expanded KIR2D+/- CD8 T cell subsets. Based on the analysis of differentially expressed genes and protein-protein interaction networks as well as the patient data, the authors conclude that KIRD2L1+ CD8+ T cells exhibit a profile of “efficient tumor immunosurveillance” while KIRD2L2+/S2+ CD8+ T cells present a profile consistent with an immune-suppressive phenotype.
Overall all, this is an interesting study which provides new insights on KIR+ CD8+ T cells in the context of solid cancers and the possible differential roles KIR+ CD8+ T cell subsets in the immune response to cancers. Although transcriptional analysis is limited by being performed on in vitro expanded KIR2D+/- CD8+ populations of healthy donors rather than directly on KIR2D+/- cells isolated form cancer patients, the extensive transcriptional characterisation, functional pathway analysis and in depth discussion provide novel and relevant insights on KIR+ CD8 T cell biology.
The authors may want to consider addressing the following points to further enhance the quality of the manuscript:
- In the introduction the authors state the interactions KIR subtypes KIRD2L1, D2L2/S2/L3 and 3DL1 are the best described. Further details on the current understanding of the role of these receptors in T or even NK cell biology in cancer settings would help put data presented in this manuscript into additional context. We have extended the introduction to describe the rest of the KIR/ligand interactions and review their role in the anti-tumor response. (lines 103-127)
- The subheading of the first results section states C2-ligand was associated with expansions of KIR2DL1+ CD8+ T cells in surviving patients. However, the data presented in Fig 2B shows that 2DL1+ CD8+ T cells are also expanded in healthy controls indicating that rather than the receptor upregulation correlating with protection, it is perhaps the loss of this receptor that correlates with poor survival? Did these patients have higher 2DL1+ CD8+ T cell frequencies at diagnosis? If the data is available it would be interesting to add. I am afraid the reviewer could have misinterpreted these data. Figure 2b shows how the presence of the C2-ligand induces the expansion of KIR2DL1+ CD8+ T lymphocytes. This occurs in healthy controls and the three types of cancer. However, the expansion induced by the C2-ligand was blocked in patients who succumbed during follow-up (Fig 2A). This suggests that these cells could be playing a relevant role in the survival of patients.
- Do KIR2DL1 and KIRD2L2+/S2+ NK cells show similar patterns of loss and expansion in non-survivors as CD8+ T cells? And if so, could the overall survival be dependent on those subsets rather than CD8 T cells? Please have a look at the first answer of the first reviewer, as the question is similar to this.
Minor comments
- Line 116-117 should read “….. and shorter patient survival” Yes, thanks.
- Line 130, the data referred to here is in Figure 2F. Yes, thanks.
- Figure 3A, third panel should read CD3- CD56+ cells. Yes, thanks.
Finally some more general thoughts that may be of interest to address in the future or include in the discussion
- Can KIR2D+ CD8 T cells be detected in the tumour microenvironment or only in the peripheral blood? We are really sorry, but we have no data in this regard. As you say, it would be very interesting to try to develop future works that allow us to analyze this aspect. Thank you for the suggestion.
- Functional assays on the ability of the 2DL1+ CD8+ T cells to either directly affect cancer cells or to support the action of cytotoxic, tumour-specific CD8 T cells or NK cells might provide further insight. Similarly, the effects of KIRD2L2+/S2+ CD8+ T cells on tumour-specific populations could be tested in in vitro co-cultures. We agree with the reviewer that these results should be validated in future functional studies. We have modified the final conclusion in the discussion to make this statement clear “In conclusion, and although a direct role of KIR+ CD8+ T cell in the containment or escape of cancer should be validated in future functional assays,…..” (lines 385-386)
Round 2
Reviewer 1 Report
Really great response from the authors - they've enhanced their paper which is a really interesting study with compelling data that is highly relevant to tumour immunology.